# Measurement of the Thermophysical Properties of Anisotropic Insulation Materials with Consideration of the Effect of Thermal Contact Resistance

**DOI:** 10.3390/ma13061353

**Published:** 2020-03-17

**Authors:** Dongxu Han, Kai Yue, Liang Cheng, Xuri Yang, Xinxin Zhang

**Affiliations:** School of Energy and Environmental Engineering, University of Science and Technology Beijing, Beijing 100083, China; handongxu131207@163.com (D.H.); clfighting@163.com (L.C.); yangxuri1207@163.com (X.Y.); zhangxx_ustb@yeah.net (X.Z.)

**Keywords:** anisotropic material, cross-plane thermal conductivity, in-plane thermal conductivity, small-plane heat source, thermal contact resistance

## Abstract

A novel method involving the effect of thermal contact resistance (TCR) was proposed using a plane heat source smaller than the measured samples for improving measurement accuracy of the simultaneous determination of in-plane and cross-plane thermal conductivities and the volumetric heat capacity of anisotropic materials. The heat transfer during the measurement process was mathematically modeled in a 3D Cartesian coordinate system. The temperature distribution inside the sample was analytically derived by applying Laplace transform and the variables separation method. A multiparameter estimation algorithm was developed on the basis of the sensitivity analysis of the parameters to simultaneously estimate the measured parameters. The correctness of the algorithm was verified by performing simulation experiments. The thermophysical parameters of insulating materials were experimentally measured using the proposed method at different temperatures and pressures. Fiber glass and ceramic insulation materials were tested at room temperature. The measured results showed that the relative error was 1.6% less than the standard value and proved the accuracy of the proposed method. The TCRs measured at different pressures were compared with those obtained using the steady-state method, and the maximum deviation was 8.5%. The thermal conductivity obtained with the contact thermal resistance was smaller than that without the thermal resistance. The measurement results for the anisotropic silica aerogels at different temperatures and pressures revealed that the thermal conductivity and thermal contact conductance increased as temperature and pressure increased.

## 1. Introduction

Anisotropic insulating materials are vastly applied in the construction industry and other related fields owing to the excellent thermal insulating properties. Accurate thermophysical parameters are imperatively needed for the effective practical application of these materials. However, the thermal contact resistance (TCR) in thermal property measurement systems is unavoidable in many cases and will affect the measurement accuracy of the thermophysical parameters of anisotropic insulation materials.

In a thermal property measurement system, TCR usually exists at interfaces between a sensor or a heating source element and the measured sample or between two samples because of the incomplete contact between two parts. The TCR can be characterized by the ratio of the temperature difference to the average heat flux density. Studies have shown that TCR is a nonlinear problem affected by multiple factors such as the roughness and topography of contact surfaces [1,2,3], external pressure loading [4,5,6], properties of intermediate media, and time-dependent temperature condition. Some theoretical models have been proposed by statistically analyzing the morphological characteristics of contact surfaces [7,8,9], temperature distributions [10], and deformation characteristics at interfaces. In practice, the accurate value of TCR is difficult to obtain through only theoretical analysis, and experimental methods are widely used to measure TCR [11,12,13,14]. According to the definition of TCR, the steady-state method was developed for TCR measurement by determining the temperatures on the contact interface and applied heat flux [15,16,17]. For transient methods, TCR is known by measuring the changes in correlated parameters under an applied thermal disturbance [18,19,20]. For instance, the intrinsic relationship between TCR and contact resistance is employed to determine TCR by measuring contact resistance [21]. The laser flash method has been proposed to measure the specific heat and TCR of a two-layer material [22]. 

Some studies have attempted to improve the accuracy of measuring the thermophysical parameters by establishing a modified measurement model involving the TCR for certain specific methods. The degree of influence of TCR on the accuracy of measuring the thermophysical parameters of materials depends on the method itself. These methods are divided into steady-state and transient methods. The TCR of ethylene tetrafluoroethylene sheets, Nafion membranes, and gas diffusion layers is measured using the hot-plate method with a more detailed consideration of TCR [23].The deviation of the measured results of ethylene tetrafluoroethylene sheets reaches 66.7% in comparison with the results measured using the traditional transient plane source method. Xu et al. [24] applied the effective medium theory to study the influence of thermal resistance at the interface between an inclusion and a substrate on the thermal conductivity of concrete. They showed that interfacial thermal resistance plays an important role in predicting the effective thermal conductivity of cement materials with coarse inclusions. Cheng et al. [25] provided a new thermal probe method by considering the thermal capacity of a thermal probe and TCR between the probe and the sample to improve the accuracy of measuring the thermal conductivity and volumetric heat capacity. Several liquid and solid samples have been measured, and the results show that TCR greatly affects the thermal conductivity of solid samples. In the measurement of the thermal diffusivity of stainless steel 304, oxygen-free copper, and aluminum nitride ceramic using the laser photothermal method, the change in TCR with temperature is measured and an empirical formula for the TCR-temperature relationship is established [26].

In our previous work, we proposed a small-plane heat source method for the simultaneous measurement of the in-plane and cross-plane thermal conductivities of anisotropic materials [27]. Given that TCR is at a contact interface between a heat source element and a sample, this work was performed to investigate a novel method in which the effect of TCR is considered to enhance the accuracy of the simultaneous measurement of thermal conductivities and volumetric heat capacity. The TCR between the heat source element and the sample in the measurement system was also determined at different pressures and temperatures.

## 2. Methods and Verification

### 2.1. Physical Mathematical Model

The measurement model with consideration of the effect of the TCR between the heating element and measured sample was established on the basis of our previously proposed small-plane heat source method, and the schematic is shown in Figure 1. As shown in Figure 1a, a plane source element with a side length of 2a was located between two samples with the same side length of 2b, i.e., samples I and II, and used as a heat source. The temperature increase at the measuring points was determined using a thermocouple with fine wires (*d* = 0.05 mm). Figure 1b shows the computational domain in the model and three spatial dimensions.

A three-dimensional mathematical model was constructed for describing the heat transfer during the measurement process when the heat capacity and thickness of the heat source element were neglected. The governing equation and the boundary condition were expressed as follows.
(1)$$\rho c\frac{\partial T(x,y,z,t)}{\partial t}=\lambda_{x}\frac{\partial^{2}T(x,y,z,t)}{\partial x^{2}}+\lambda_{y}\frac{\partial^{2}T(x,y,z,t)}{\partial y^{2}}\lambda_{z}\frac{\partial^{2}T(x,y,z,t)}{\partial z^{2}}$$
where *ρ* and *c* are the density and specific heat capacity of the sample; *T*(*x*, *y*, *z*, *t*) is the temperature of the sample at different times; and λ*_x_*, λ*_y_*, and λ*_z_* are the thermal conductivities of the measured sample in the *x*, *y*, and *z* directions, respectively.

Given that the increase in the temperature of the sample during measurement did not remarkably increase the thermal conductivity, the specific heat capacity and density of the sample were considered constant with respect to temperature. The dimensions of the sample were supposed to be large enough to satisfy the requirements for the assumption of semi-definite medium. Therefore, the upper boundary plane *z* = *b* was set as the isothermal condition at room temperature *T*_0_. Given the symmetry of the measurement model, the left boundary plane *x* = 0 and front boundary plane y = 0 were set as adiabatic boundary conditions. The temperatures of the right boundary plane *x* = *b* and back boundary plane *y* = *b* were not affected by the heating element during the experiment under the control of the heating power. Then, the right boundary plane *x* = *b* and back boundary plane *y* = *b* were also set as the adiabatic boundary conditions. The boundary condition of the lower boundary plane *z* = 0 was expressed as follows:(2)$$q=h_{c}(T_{h}-T)|{}_{z=0}=-\lambda_{z}\frac{\partial T(x,y,0,t)}{\partial z}$$
where *q* is half of the heat flux density; *h_c_* is the thermal contact conductance (TCC) between the heating element and the sample which is the reciprocal of the TCR; and *T* is the temperature at the measuring point.

The initial condition was set as room temperature *T*_0_. The sample size should be sufficiently larger than the penetration depth of heat flux to ensure that the heat flux has no thermal effect on the *z* = *b* condition plane of the semidefinite medium. 

### 2.2. Analytical Solution

First, the governing equation and boundary conditions were transformed with the Laplace transform L(ΔT(x,y,z,t))=θ(x,y,z,p) and could be written as follows:(3)$$\rho c_{p}\theta (x,y,z,p)=\lambda_{x}\frac{\partial^{2}\theta (x,y,z,p)}{\partial x^{2}}+\lambda_{y}\frac{\partial^{2}\theta (x,y,z,p)}{\partial y^{2}}+\lambda_{z}\frac{\partial^{2}\theta (x,y,z,p)}{\partial z^{2}}$$
(4)$$\frac{q}{p}=h_{c}(\theta_{h}-\theta )|{}_{z=0}=-\lambda_{z}\frac{\partial \theta (x,y,0,p)}{\partial z}$$
where p is the Laplace parameter of t; and θ is the Laplace transform of the temperature increase. θ(x,y,z,p) could be transformed into X(x,p)Y(y,p)Z(z,p) by applying the variable separation method.

Then, the temperature increase was analytically obtained as follows:(5)$$\theta (x,y,z,p)=\sum_{m=1}^{\infty }\sum_{n=1}^{\infty }B_{mn}\cos(\alpha_{m}x)\cos(\delta_{n}y)\exp(-A_{mn}z)$$

The *A_mn_* and *B_mn_* were calculated in terms of the boundary conditions as follows:(6)$$A_{mn}=\sqrt{\frac{\rho cp+\lambda_{x}\alpha_{m}^{2}+\lambda_{y}\delta_{n}^{2}}{\lambda_{z}}}$$
(7)$$B_{mn}=\frac{4\phi_{0}\left(\sqrt{\lambda_{z}\rho cp}-h\right)}{ph\sqrt{\lambda_{z}\rho cp}A_{mn}b^{2}}\frac{\sin(\alpha_{m}\alpha )}{\alpha_{m}}\frac{\sin(\delta_{n}\alpha )}{\delta_{n}}$$

The inverse Laplace transform was applied to obtain the analytical solution of the temperature increase in the time domain which is derived as:
(8)$$\begin{array}{l} \Delta T(x.y.z.t)=\sum_{m=1}^{\infty }\sum_{n=1}^{\infty }\frac{4\phi_{0}}{b^{2}\sqrt{h}}\frac{\sin(\alpha_{m}a)}{\alpha_{m}}\frac{\sin(\delta_{n}a)}{\delta_{n}}\cos(\alpha_{m}x)\cos(\delta_{n}y)\frac{1}{\sqrt{a_{x}\alpha_{m}^{2}+a_{y}\delta_{n}^{2}}}\\ \frac{1}{\sqrt{\lambda_{z}\rho c_{p}}}\left(\begin{array}{l} \text{exp}(-z\sqrt{\frac{\rho c_{p}}{\lambda_{z}}}\sqrt{a_{x}\alpha_{m}^{2}+a_{y}\alpha_{n}^{2}})erfc(\frac{z\sqrt{\frac{\rho c_{p}}{\lambda_{z}}}}{2\sqrt{t}}-\sqrt{a_{x}\alpha_{m}^{2}+a_{y}\alpha_{n}^{2}}\sqrt{t})-\\ \exp(z\sqrt{\frac{\rho c_{p}}{\lambda_{z}}}\sqrt{a_{x}\alpha_{m}^{2}+a_{y}\alpha_{n}^{2}})erfc(\frac{z\sqrt{\frac{\rho c_{p}}{\lambda_{z}}}}{2\sqrt{t}}+\sqrt{a_{x}\alpha_{m}^{2}+a_{y}\alpha_{n}^{2}}\sqrt{t}) \end{array}\right)\\ +\sum_{m=1}^{\infty }\frac{2\phi_{0}}{b^{2}\sqrt{h}}\frac{\sin(\alpha_{m}a)}{\alpha_{m}}\cos(\alpha_{m}x)\\ \frac{1}{\sqrt{\lambda_{z}\rho c_{p}}\sqrt{a_{x}\alpha_{m}^{2}}}\left(\begin{array}{l} \text{exp}(-z\sqrt{\frac{\rho c_{p}}{\lambda_{z}}}\sqrt{a_{x}\alpha_{m}^{2}})erfc(\frac{z\sqrt{\frac{\rho c_{p}}{\lambda_{z}}}}{2\sqrt{t}}-\sqrt{a_{x}\alpha_{m}^{2}}\sqrt{t})-\\ \exp(z\sqrt{\frac{\rho c_{p}}{\lambda_{z}}}\sqrt{a_{x}\alpha_{m}^{2}})erfc(\frac{z\sqrt{\frac{\rho c_{p}}{\lambda_{z}}}}{2\sqrt{t}}+\sqrt{a_{x}\alpha_{m}^{2}}\sqrt{t}) \end{array}\right)\\ +\sum_{m=1}^{\infty }\frac{2\phi_{0}}{b^{2}\sqrt{h}}\frac{\sin(\delta_{n}a)}{\delta_{n}}\cos(\delta_{n}y)\\ \frac{1}{\sqrt{\lambda_{z}\rho c_{p}}\sqrt{a_{y}\delta_{n}^{2}}}\left(\begin{array}{l} \text{exp}(-z\sqrt{\frac{\rho c_{p}}{\lambda_{z}}}\sqrt{a_{y}\alpha_{n}^{2}})erfc(\frac{z\sqrt{\frac{\rho c_{p}}{\lambda_{z}}}}{2\sqrt{t}}-\sqrt{a_{y}\alpha_{n}^{2}}\sqrt{t})-\\ \exp(z\sqrt{\frac{\rho c_{p}}{\lambda_{z}}}\sqrt{a_{y}\alpha_{n}^{2}})erfc(\frac{z\sqrt{\frac{\rho c_{p}}{\lambda_{z}}}}{2\sqrt{t}}+\sqrt{a_{y}\alpha_{n}^{2}}\sqrt{t}) \end{array}\right)\\ +\frac{2a^{2}\phi_{0}}{b^{2}\sqrt{\lambda_{z}h\rho c_{p}}}\left(\sqrt{\frac{4t}{\pi }}\exp(-\frac{z^{2}\rho c_{p}}{4t\lambda_{z}})-z\sqrt{\frac{\rho c_{p}}{\lambda_{z}}}erfc(z\sqrt{\frac{\rho c_{p}}{4t\lambda_{z}}})\right) \end{array}$$
where $\alpha_{x}$ and $\alpha_{y}$ are the thermal diffusion coefficient in the x-direction and y-direction, respectively; $\alpha_{m}=\frac{m\pi }{b}(m=1,2,3\ldots );and\;\delta_{n}=\frac{n\pi }{b}(n=1,2,3\ldots )$.

The thermal conductivities and volumetric heat capacity of the sample and the TCC could be accurately estimated using Equation (8) and appropriate parameter estimation methods. The analytical solution for isotropic materials could be obtained by setting $\lambda_{x}=\lambda_{y}=\lambda_{z}$ into Equation (8). Given the reaction time of the thermocouple, the temperature measured at time *t* should be the one obtained at time *t−t_d_*. Substituting *t = t − t_d_* into Equation (8), the analytical solution with the reaction time for the temperature of anisotropic materials could be achieved. Herein, the reaction time should satisfy $0\leq t_{d}\leq 0.005t$, and *t* is the total test time [28].

### 2.3. Comparison of Analytical and Numerical Solutions

The numerical simulation of the heat transfer of this measurement model was performed using FLUENT 16.0. The numerical simulation results of the temperature increase at the measuring point (0,0,0.005) for isotropic and anisotropic materials were compared with those obtained from the analytical solution. Table 1 shows the thermal–physical properties used for the calculation.

Figure 2 shows the comparative results between the numerical and analytical solutions for isotropic (see Figure 2a) and anisotropic (see Figure 2b) materials at the measuring point (0,0,0.005). The maximum deviation in the temperature increase for the two materials was less than 1.6%, indicating the correctness of the analytical solution in this study.

### 2.4. Simultaneous Estimation of Thermophysical Parameters

Firstly, the sensitivity analysis was carried out and the sensitivity coefficients were calculated as follows to relate the magnitude of the thermal parameters to the change in the temperature increase at the measuring point. The sensitivity coefficient for the cross-plane thermal conductivity was obtained by $X_{ij}^{+}=\alpha_{j}X_{j}(t_{i},\alpha )=\alpha_{j}\frac{\partial T(t_{i},\alpha )}{\partial \alpha_{j}}$, where $X_{ij}^{+}$ represents the sensitivity coefficient of an estimated parameter $\alpha_{j}$ (i.e., $\lambda_{xy}$, or $\lambda_{z}$, or $\rho C_{p}$, or $h_{c}$) to the temperature $T(t_{i},\alpha )$. These coefficients were also used in the inverse problem solving to identify the parameters. Using the values listed in Table 1, the results of sensitivity coefficients calculation for the estimated thermophysical parameters and TCC are shown in Figure 3.

As can be seen in Figure 3, the curves of the sensitivity coefficients of four estimated parameters were all nonlinear and uncorrelated to each other. It was indicated that the sensitivity coefficients of the estimated parameters were linearly independent of one another, and there was no set of correlated parameters. According to the basic principle of parameter estimation [29], the thermal conductivities $\lambda_{xy}$ and $\lambda_{z}$, volumetric heat capacity, and TCC could be estimated simultaneously.

Then, an improved Gauss–Newton algorithm was developed to simultaneously obtain four estimated parameters [30] by the least squares analysis in which the sum of the square errors between the calculated and experimental results of the temperature increase of the measuring point in the sample was minimized. A series of simulation experiments were performed to verify the correctness of the method for estimating multiple parameters and the stability of the estimation results. The temperature signal obtained by the analytical solution was added with different random noises to be used as the experimental measurement data. The signal can be expressed as $\Delta T=\Delta T_{0}[1+(\varepsilon r\;and-\varepsilon /2)]$, where ε is the value of random error,$\Delta T_{0}$ is the temperature increase obtained by the analytical solution at the measuring point, and *rand* is used to generate random matrices. In the simulation experiments, the in-plane and cross-plane thermal conductivities of the anisotropic materials were set to 0.4 W∙m^−1^·K^−1^ and 0.2 W∙m^−1^·K^−1^, and the volumetric heat capacity was set to 1,500,000 J∙m^−3^·K^−1^. The values of other parameters were the same as those listed in Table 1 for the isotropic materials. Figure 4 shows the comparison of the increase in temperature between the simulated and estimated results as well as the residual errors for isotropic and anisotropic materials.

It was seen that the estimated and simulated results of the increase in temperature agreed well, and the residual errors were small. All the estimated errors of the estimated parameters of the isotropic and anisotropic materials were less than 2.8% at different random errors. This result verified the feasibility and effectivity of the developed method for the simultaneous estimation of the in-plane and cross-plane thermal conductivities, volumetric heat capacity, and TCC based on the proposed small-plane heat source method with TCR.

## 3. Experimental Measurement

### 3.1. Experimental System

Figure 5 shows the measurement system established according to the proposed method. As can be seen, the system mainly included a power supply, a computer-based data acquisition system, a pressure loading device (see Figure 5a), a Cr20Ni80 alloy heating element with a size of 40 mm × 40 mm × 0.1 mm, and a resistance of 10.2 Ω (see Figure 5b), and the tested samples consisting of three squares of the same size but different thicknesses (see Figure 5c, i.e., samples A, B, C, and D). The constant-temperature muffle furnace shown in Figure 5d is a box-type resistance furnace with an accuracy of ±1 °C (SRJX-8-13, Tianjin, China) and was used for the experimental measurements performed at different high temperatures. The power supply (PA36-2A, KENWOOD, Akaho, Japan) provided a constant current output for heating. The multi-channel data acquisition instrument (MX-100, YOKOGA, Tokyo, Japan) was used to record the temperature rise at the measuring point of the sample which was detected by a K-type thermocouple with a diameter of 0.05 mm (OMEGA, Norwalk, CT, USA).

### 3.2. Experimental Verification

First, the thermophysical parameters of two kinds of standard reference materials (SRMs), including a glass fiber insulating material (Sample A) and ceramic material (Sample B) produced by the Institute of Standardization of National Building Materials (China), were measured using the proposed method with consideration of the effect of TCR to validate its feasibility and accuracy. The standard value of cross-plane thermal conductivity of sample A was 0.0331 ± 0.0002 W∙m^−1^∙K^−1^, which was obtained by the guarded hot-plate method at 28 °C. The standard value of the cross-plane thermal conductivity of Sample B was 0.052 ± 0.0002 W∙m^−1^∙K^−1^ as obtained by the heat flow meter. Table 2 shows the experimental results of the Samples A and B.

As can be seen, the relative measurement errors of Samples A and B were 0.91% and 1.53% in comparison with the standard values, respectively. Thus, it was concluded that the established measurement system could be used to measure the thermal properties with good accuracy.

Then, the TCR of the insulation tile material (Sample C, ceramic insulation tiles) were measured using the traditional steady-state method [17] and the proposed method at different pressures and temperatures. The measured TCCs are shown in Table 3.

The TCCs obtained by the proposed method increased as the temperature and pressure increased. This variation in the TCC was rational and in agreement with that measured with the steady-state method. The values of the TCCs measured with the proposed method were smaller than those obtained by the steady-state method, and the deviation was within 8.5%. This phenomenon might be due to the difficulty in completely satisfying the requirement of 1D heat transfer in the practical measurement system for the steady-state method, and heat was loss through the thermocouples and through radiation and convection. 

### 3.3. Experimental Measurement at High Temperatures

The thermophysical parameters and the TCCs of an anisotropic high-temperature-resistant material (Sample D, silica aerogel material) with a large difference in its in-plane and cross-plane thermal conductivities were experimentally measured at room and high temperatures by using the proposed method. For comparison, the same thermophysical parameters of the same sample were measured using the small-plane heat source method without the effect of TCR which was reported in our previous work [27]. The corresponding changes with the increase in temperature are shown in Figure 6.

It was indicated that an increase in temperature led to an increase in thermal conductivities, volumetric heat capacity, and TCC. The cross-plane and in-plane thermal conductivities and volumetric heat capacity increased by 20.2%, 37.9%, and 12.3% when the temperature increased to 800 °C compared with those at 28 °C, respectively. The TCC of the material also increased by 21.7%. This result indicated that the thermophysical parameters of the material and the TCR of the measurement system might greatly change with the temperature. The change in the in-plane thermal conductivity was larger than that in the cross-plane thermal conductivity. The thermal conductivities measured by the method with the effect of the TCR were less than those obtained by the method without the TCR. The maximal deviation was 8.21% when the TCC between the sample and the heating element was 125.1 W·m^−1^·K^−1^, because the increase in the temperature of the sample calculated by the mathematical model of the measurement system would be larger than that measured by the experimental measurement if the TCR was not considered. Hence, leading the measured value of the thermal conductivities estimated using the experimental data was larger than the desired value. In addition, the in-plane thermal conductivity showed higher difference between the TCR consideration and without the TCR consideration as the temperature increased. A possible reason might be that the change with temperature in the in-plane thermal conductivity was larger than that in the cross-plane thermal conductivity, and, correspondingly, the thermal resistance in the in-plane direction was relatively smaller. Subsequently, the effect of the TCR on the heat transfer in the in-plane direction was larger than that in the cross-plane direction in response to increasing temperature.

The uncertainty in experimental results was mainly affected by the temperature measurement, the determination of the power supply, the location of the measuring point, and the time delay. In this study, the uncertainties caused by temperature measurement, time counting, and location of the measuring point were 0.43%, 0.2%, and 0.3%, respectively. The uncertainty of output power from the power supply was 0.2%. The maximum difference in temperature of the results from parameter estimation between the experimental measurement and theoretical calculation was less than 0.06 °C, and the average residual error was less than 0.04 °C. The relative uncertainty was 2.0% based on the absolute average value of the temperature increase. Thus, the overall uncertainty of the measurement system was 2.3%.

## 4. Conclusions

In this study, the small-plane heat source method was improved by considering the effect of TCR on the thermophysical parameter measurement. The heat transfer process in the measurement was theoretically modeled and analytically solved. A multiparameter estimation method was developed on the basis of sensitivity analysis to simultaneously measure the desired thermophysical parameters and TCR. The thermophysical parameters of different samples and TCR in the system were subjected to simulation and experimental measurements at different temperatures and pressures to validate the proposed method. The comparison of thermal conductivity between the measured value and the standard value showed that the relative error was less than 2%, proving the correctness of the proposed method. The measured result with the effect of TCR was smaller than that without this effect, indicating that TCR should be considered. Uncertainty analysis showed that the uncertainty of the whole system was 2.3% which could meet the requirement of an experimental measurement. The proposed method could be used to simultaneously determine the in-plane and cross-plane thermal conductivities, volumetric heat capacity, and TCC with high measurement accuracy for various anisotropic materials.

## Figures and Tables

**Figure 1 materials-13-01353-f001:**
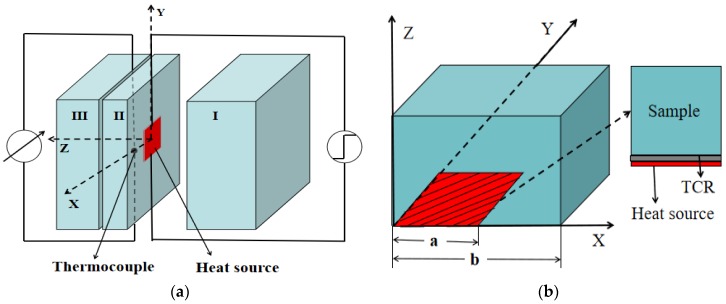
Schematic diagram: (**a**) experimental system; (**b**) the modeled sample and location of thermal contact resistance (TCR).

**Figure 2 materials-13-01353-f002:**
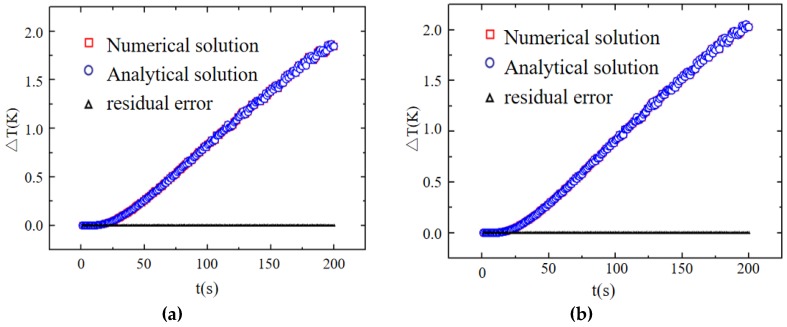
Comparison between the numerical and analytical solutions: (**a**) isotropic and (**b**) anisotropic materials.

**Figure 3 materials-13-01353-f003:**
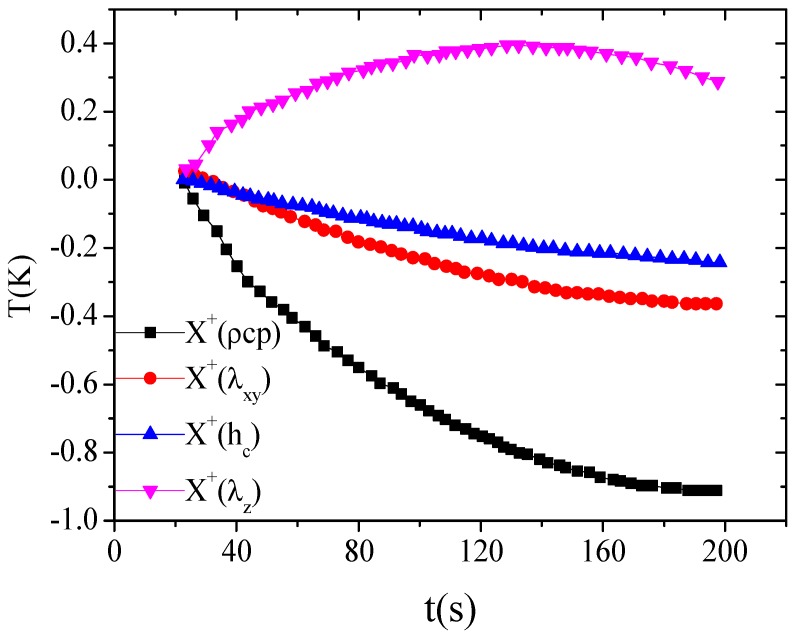
Results of the parameter sensitivity analysis.

**Figure 4 materials-13-01353-f004:**
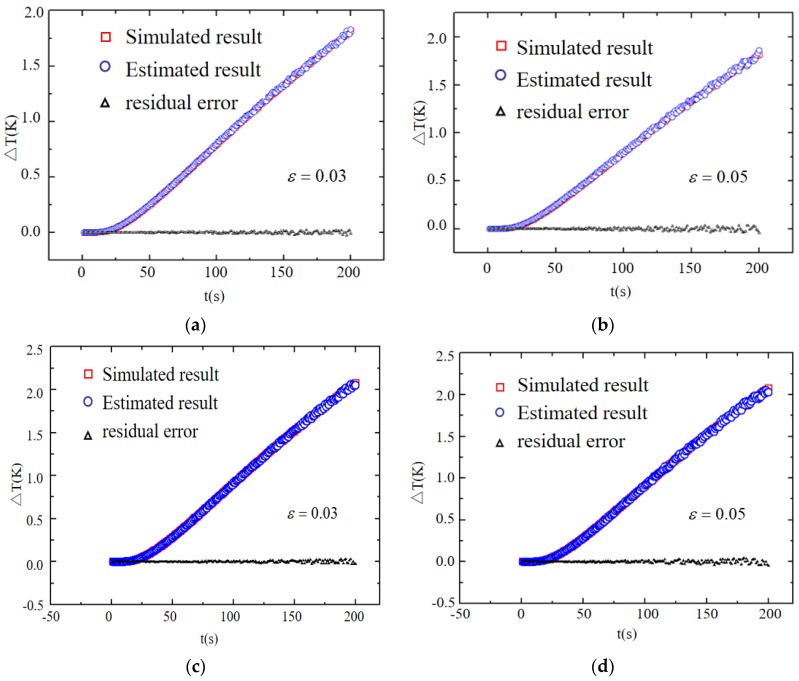
Comparison of the increase in temperature between the simulated and estimated results: (**a**) isotropic material, ε = 0.03; (**b**) isotropic material, ε = 0.05; (**c**) anisotropic material, ε = 0.03; and (**d**) anisotropic material, ε = 0.05.

**Figure 5 materials-13-01353-f005:**
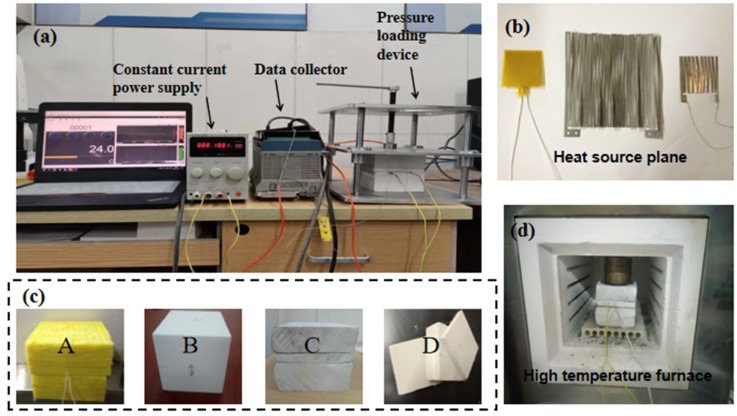
Schematic of the measurement system. (**a**) experimental system; (**b**) heat source plane; (**c**) samples; (**d**) high temperature furnace.

**Figure 6 materials-13-01353-f006:**
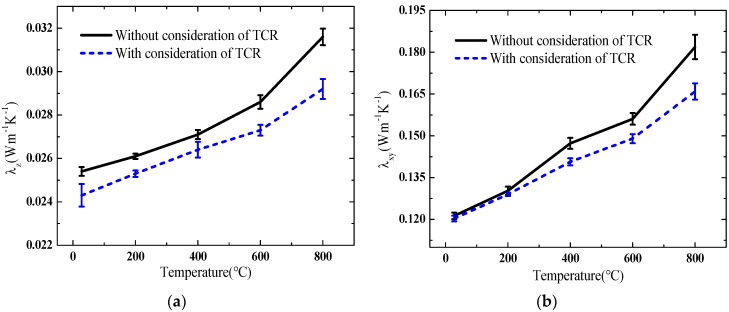
Results of experimental measurement at different temperatures: (**a**) cross-plane thermal conductivity; (**b**) in-plane thermal conductivity; (**c**) TCC; and (**d**) volumetric heat capacity.

**Table 1 materials-13-01353-t001:** Thermal–physical properties used in numerical simulations.

Parameters	Isotropic Material	Anisotropic Materials
*λ_z_* (W·m^−1^·K^−1^)	0.4	0.2
*λx* = *λ_y_* (W·m^−1^·K^−1^)	0.4	0.4
*q* (W·m^−2^)	80	80
*b* (m)	0.05	0.05
*ρc_p_* (J·m^−3^·k^−1^)	1,599,750	1,500,000
a (m)	0.02	0.02
t (s)	200	200
*h_c_* (W·m^−2^·K^−1^)	150	150
Coordinates (*x*,*y*,*z*) (m)	(0,0,0.005)	(0,0,0.005)

**Table 2 materials-13-01353-t002:** Results of verification experiments.

Sample	$\lambda_{z}\pm SD$ (W·m^−1^·K^−1^)	$\lambda_{xy}\pm SD$ (W·m^−1^·K^−1^)	h±SD (W·m^−2^·K^−1^)	$\rho C_{P}\pm SD$ (J∙m^−3^∙K^−1^)
A	0.0334 ± 0.0001	0.0452 ± 0.0004	173.0 ± 1.665	1.595 ± 0.0011
B	0.0528 ± 0.0012	0.156 ± 0.00473	122.9 ± 1.967	3.033 ± 0.0484

**Table 3 materials-13-01353-t003:** Experimental results of thermal contact conductivity using the proposed methods and the steady state.

T (°C)	P (kPa)	Thermal Contact Conductivity (W·m^−2^·K^−1^) ± SD		
		Proposed Method	Steady-State Method	Deviation (%)
200	2	94.1 ± 0.35	102.1 ± 0.45	8.5
	5	98.5 ± 0.42	105.8 ± 0.36	7.4
	8	102.4 ± 0.25	110.7 ± 0.21	8.1
300	2	97.3 ± 0.46	104.4 ± 0.71	7.3
	5	101.9 ± 0.42	108.8 ± 0.43	6.8
	8	104.8 ± 0.38	112.7 ± 0.58	7.5
400	2	99.2 ± 0.57	105.5 ± 0.65	6.4
	5	105.7 ± 0.55	112.9 ± 0.54	6.8
	8	108.9 ± 0.43	116.6 ± 0.32	7.1

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
