# Peer review of "Measurement of the Thermophysical Properties of Anisotropic Insulation Materials with Consideration of the Effect of Thermal Contact Resistance"

_materials, 2020, doi:10.3390/ma13061353_

Round 1

Reviewer 1 Report

In this paper, the author shows the thermophysical properties of anisotropic insulation material (silica aerogel) with consideration of the effect of thermal contact resistance. I find value of volumetric heat capacities are different in table I and in page 8 line 175. Please check. In this paper the author used conductivity of Isotropic material and conductivity of anisotropic material 0.4(W/mK) and 0.2(W/mK) respectively. Please explain the reason why, and provide the basic dada of the silica aerogel used in this experiment. In Figure 6 (b), as increase temperature the in-plane thermal conductivity shows higher different between with the TCR consideration and without the TCR consideration. And it should be discussed as well. The author needs to be update in page 10, “3.1. Experimental verification” to “3.2. Experimental verification”, in page 13, “3.Conclusion” to “4.Conclusion” and in page 14, “[26]” to “[27]”. The most serious problem of the manuscript consists the duplicate data (for example table 2 and figure 4, table 5 and figure 6), it should show clear data in this work.

Reviewer 2 Report

   The manuscript improves the small-plane heat method of Ref. [27], by considering the effect of thermal contact resistance on the thermophysical parameter measurement. The method is validated using various simulations at different temperatures and pressure, found a relative error less than 2 %. The manuscript is interesting and it can be published in Materials journal.

   I have some suggestions:

   -a critical assessment of the method can be also important. When the method is not accurate? When the variable separation technique, used by the authors, is not realistic? Can be found an example where numerical and analytical solution are not in such good agreement?

   -Fig. 3 should be better explained and all sensitivity coefficients must be defined

   - abstract: TCR not defined
   - the format of the Table below Keywords is not correct, same for Table 2

Round 2

Reviewer 1 Report

The manuscript is showing properly revised.

I recommend to address minor revisions before for the publication in Materials.

1. Please explain more detail about figures in Fig. 5 in page 9.

2. Please separate “(a) (b)” in figure 1.

3. Please use same functions in equation (8) (“ Erfc” or “erfc”) in page 5.
